# Bidirectional ATP-driven transport of cobalamin by the mycobacterial ABC transporter BacA

Mark Nijland[1], Solène N. Lefebvre[1,3], Chancievan Thangaratnarajah [1,2,3] & Dirk J. Slotboom [1] ✉

BacA is a mycobacterial ATP-binding cassette (ABC) transporter involved in the translocation of water-soluble compounds across the lipid bilayer. Whole-cell-based assays have shown that BacA imports cobalamin as well as unrelated hydrophilic compounds such as the antibiotic bleomycin and the antimicrobial peptide Bac7 into the cytoplasm. Surprisingly, there are indications that BacA also mediates the export of different antibacterial compounds, which is difficult to reconcile with the notion that ABC transporters generally operate in a strictly unidirectional manner. Here we resolve this conundrum by developing a fluorescence-based transport assay to monitor the transport of cobalamin across liposomal membranes. We find that BacA transports cobalamin in both the import and export direction. This highly unusual bidirectionality suggests that BacA is mechanistically distinct from other ABC transporters and facilitates ATP-driven diffusion, a function that may be important for the evolvability of specific transporters, and may bring competitive advantages to microbial communities.

ATP-binding cassette (ABC) transporters comprise a large family of primary active transporters found in all domains of life that mediate the ATP-driven translocation of a wide range of substrates into or out of the cell or organelles. They are generally considered to be unidirectional transporters in physiological conditions, acting as either importer or exporter[1–7], with the notable exception of CFTR that functions as an ATP-gated chloride channel[8]. The cytosolic nucleotide-binding domains (NBDs) of ABC transporters are highly conserved and represent the hallmark of the transporter family, whereas transmembrane domains (TMDs) that form the translocation pathway are more diverse among the members. A recent classification of ABC transporters based on the folding of these more variable TMDs placed all known members into seven distinct groups (type I–VII)[2]. Type I–III ABC transporters act strictly as importers and are found exclusively in prokaryotes, while the type IV–VII mostly include export systems[2]. Type IV ABC transporters represent an interesting class in which most

members have an export function, but some transporters have recently been shown to be importers, including Rv1819c from *Mycobacterium tuberculosis* (also referred to as BacA[Mt])[9,10].

A genetic screen using random transposon insertions revealed that BacA[Mt] is the sole transporter responsible for the import of cobalamin (vitamin B12) in *M. tuberculosis*[11]. This import may be essential, because unlike other mycobacteria, *M. tuberculosis* does not appear to have the ability to synthesize the micronutrient de novo[12]. The protein is important to maintain chronic infections in mice[13]. Additionally, BacA[Mt] has been implicated in the import of other structurally unrelated hydrophilic compounds, including several antimicrobial compounds, such as the proline-rich peptide Bac7[(1-16)][13,14], bleomycin[9,11,13], evybactin[15] and aminoglycosides such as streptomycin[16]. As it is unlikely that BacA[Mt] has evolved for the specific import of these toxic compounds, it is assumed that they are taken up into the cell as part of a moonlighting activity of the protein. The

[1]Faculty of Science and Engineering, Groningen, Biomolecular Sciences and Biotechnology, Membrane Enzymology Group, University of Groningen, 9747 AG Groningen, The Netherlands. [2]Sosei Heptares, Steinmetz Building, Granta Park, Great Abington, Cambridge CB21 6DG, UK. [3]These authors contributed equally: Solène N. Lefebvre, Chancievan Thangaratnarajah. ✉e-mail: d.j.slotboom@rug.nl

relevance of the observed polyspecificity or promiscuity is poorly understood. A recently solved cryo-EM structure of BacA$_{Mt}$ in the nucleotide-bound occluded state has shed some light on its polyspecificity[9]. The protein possesses a large internal cavity of ~7700 Å$^3$ that spans the bilayer thickness. The cavity's surface is lined with negatively charged and polar residues, and appears to lack an obvious high-affinity binding site for its substrates. The cavity instead provides a generally favorable environment for the accommodation of hydrophilic compounds, which may explain how structurally unrelated compounds can be transported.

Although it is generally viewed that ABC transporters act as unidirectional transporters, it has also been speculated that BacA$_{Mt}$ not only imports substrates, but may also function as an exporter for other substrates. Several whole-cell-based studies have shown that the treatment of drug-resistant clinical isolates of *M. tuberculosis* with rifampicin and isoniazid is associated with mutations or increased expression levels of BacA$_{Mt}$[17–23], which is a common observation for drug-efflux pumps[24]. Accordingly, heterologous expression of BacA$_{Mt}$ in *M. smegmatis* increases its resistance to rifampicin, further supporting its potential role in the export of the drugs from the cell[23]. BacA$_{Mt}$ is also essential for *M. tuberculosis* to protect itself against the cysteine-rich host peptide human β-defensin (HBD2), as the bacterium becomes more susceptible to the peptide upon the deletion of *bacA$_{Mt}$*[14]. At the same time, its heterologous expression increases the resistance of *Sinorhizobium meliloti* to HBD2 and another cysteine-rich peptide NCR247[14], suggesting that BacA$_{Mt}$ may also actively transports these peptides out of the cell. It is noteworthy that the characterization of the substrate transport processes mediated by BacA$_{Mt}$ has so far been limited to whole-cell-based assays, which are indirect, sometimes difficult to interpret, and complicate mechanistic interpretation of the import and export functions.

In this work, we establish a robust in vitro fluorescence-based transport assay for cobalamin to measure the transport activity of purified and liposome-reconstituted protein. Using BacA from *Mycobacterium marinum* (BacA$_{Mm}$, or MMAR2696, 88.2% sequence identity with BacA$_{Mt}$, Supplementary Fig. 5), we demonstrate that the transport system indeed operates in the import direction for the substrate cobalamin, as it had been shown previously in whole-cell assays[9,11]. Unexpectedly, we show that BacA$_{Mm}$ also mediates ATP-driven transport of cobalamin in the export direction, demonstrating that a member of the ABC transport family acts as a bidirectional transporter for a single physiological substrate. This work shows that the mechanistic diversity of ABC transporters is larger than previously thought, and that functions typically associated with secondary transporters can also be incorporated in an ABC transporter.

## Results

### Fluorescence-based transport assay for cobalamin

We used the previously engineered single-cysteine variant D141C of the cobalamin-specific substrate binding protein (SBP) BtuF from *Escherichia coli* to generate a fluorescent sensor for cobalamin[25]. We purified the protein and labelled the cysteine with a maleimide version of fluorophore Alexa Fluor 488 yielding the chemically modified protein BtuF$_{488}$ (Supplementary Fig. 1a). Binding of cobalamin to BtuF$_{488}$ brings the ligand in close proximity to the conjugated fluorophore, which has previously been shown to quench the fluorescence signal by 76%[25]. To establish the usefulness of the sensor for high-throughput transport measurements in a 96-well plate format, we titrated cobalamin to the sensor in solution and monitored the fluorescence change. We observed a substrate concentration dependent quenching to a maximum of nearly 75% and calculated a dissociation constant ($K_d$) of BtuF$_{488}$ for cobalamin of 11.5 nM (Fig. 1a), which is similar to the value reported previously[26,27].

Next, we envisaged that BtuF$_{488}$ could be used as sensor to monitor transport of cobalamin across liposomal membranes (Supplementary Fig. 4). To develop a transport assay, we purified and reconstituted the energy-coupling factor (ECF) transporter ECF-CbrT from *Lactobacillus delbrueckii* into liposomes (Supplementary Fig. 1b). ECF-CbrT is a Type III ABC transporter that mediates ATP-dependent import of cobalamin in bacteria[28,29]. We define the side of the membrane where the NBDs are located as *cis*-side, and the opposite side the *trans*-side. ECF-CbrT functions as an importer and therefore transports cobalamin from the *trans* to the *cis* side. The reconstitution procedure of ECF transporters has been shown to result in both inside-out and right-side-out oriented protein complexes in the liposomal membrane[30], where the transporters expose their cytosolic NBDs to the external buffer or to the aqueous solution within the liposomal lumen, respectively. To monitor the transport activity of only one of these two populations of ECF-CbrT, we added Mg-ATP and cobalamin to the buffers on opposite sides of the membrane, and the sensor to the same compartment as Mg-ATP (Fig. 1b). To probe the activity of the population exposing their NBDs (*cis*-side) to the external buffer, we encapsulated cobalamin in the lumen of the proteoliposomes (*trans*-side), and added the sensor to the external buffer. The transport reaction was subsequently started by adding Mg-ATP to the external buffer and we followed the fluorescence over time in a 96-well plate. While *trans*-to-*cis* transport physiologically leads to import of cobalamin, in our liposome set-up it leads to release of the substrate from the lumen (physiological import direction). A time-dependent quenching of the sensor fluorescence was observed in the presence of Mg-ATP, indicating that cobalamin is actively transported out of the proteoliposomes (yellow circles, Fig. 1b). The fluorescence remained unaffected when Mg-ATP was replaced by Mg-ADP (white triangles, Fig. 1b), or when liposomes were reconstituted with the folate-specific ECF transporter ECF-FolT2 (yellow and grey triangles, Fig. 1b, Supplementary Fig. 1c), confirming that the observed quenching is derived from ATP-dependent transport of cobalamin across the membrane by ECF-CbrT. Next, we co-encapsulated Mg-ATP together with cobalamin in the lumen of the proteoliposomes, while keeping the sensor on the outside. In this configuration, potential transport from the *cis* to the *trans* side can be detected (physiological export direction) by the population of ECF-CbrT exposing their NBDs within the lumen of the liposomes. As expected for an importer, we did not observe quenching of the sensor (blue circles, Fig. 1b), confirming that ECF-CbrT lacks the ability to transport cobalamin in the export direction. To confirm that ECF-CbrT only transports cobalamin in the import direction, we reversed the transport setup and examined the transport activity by monitoring the accumulation of cobalamin into the lumen of proteoliposomes (Supplementary Fig. 4). To do so, we encapsulated the sensor within the lumen of the proteoliposomes, added cobalamin to the external buffer, and had Mg-ATP present either in the lumen, or in the external solution. Although this set-up leads to slightly noisier traces (since the amount of sensor molecules that can be incorporated in the lumen is limited), the results are clear (Fig. 1c). Quenching of the sensor was observed only when Mg-ATP was co-encapsulated in the lumen of the liposomes (yellow circles, Fig. 1c). In contrast, when Mg-ATP was added to the external buffer (blue circles, Fig. 1c), no quenching was observed confirming that ECF-CbrT transports cobalamin only in the import direction.

### Bidirectional transport of cobalamin by BacA$_{Mm}$

We then turned to the mycobacterial type IV ABC transporter BacA. Previously, it was shown that the cobalamin auxotrophic strain *E. coli* ΔFEC requires the expression of a functional cobalamin transporter for growth[28,31]. By monitoring its growth, it was concluded that BacA$_{Mt}$ mediates the import of cobalamin[9]. We used the same assay to show that expression of the closely-related homologue BacA from *M. marinum* (BacA$_{Mm}$, 88.2% sequence identity with BacA$_{Mt}$) also rescues growth of the *E. coli* ΔFEC strain (Supplementary Fig. 2a), showing that BacA$_{Mm}$ is also a functional cobalamin importer. For experiments with

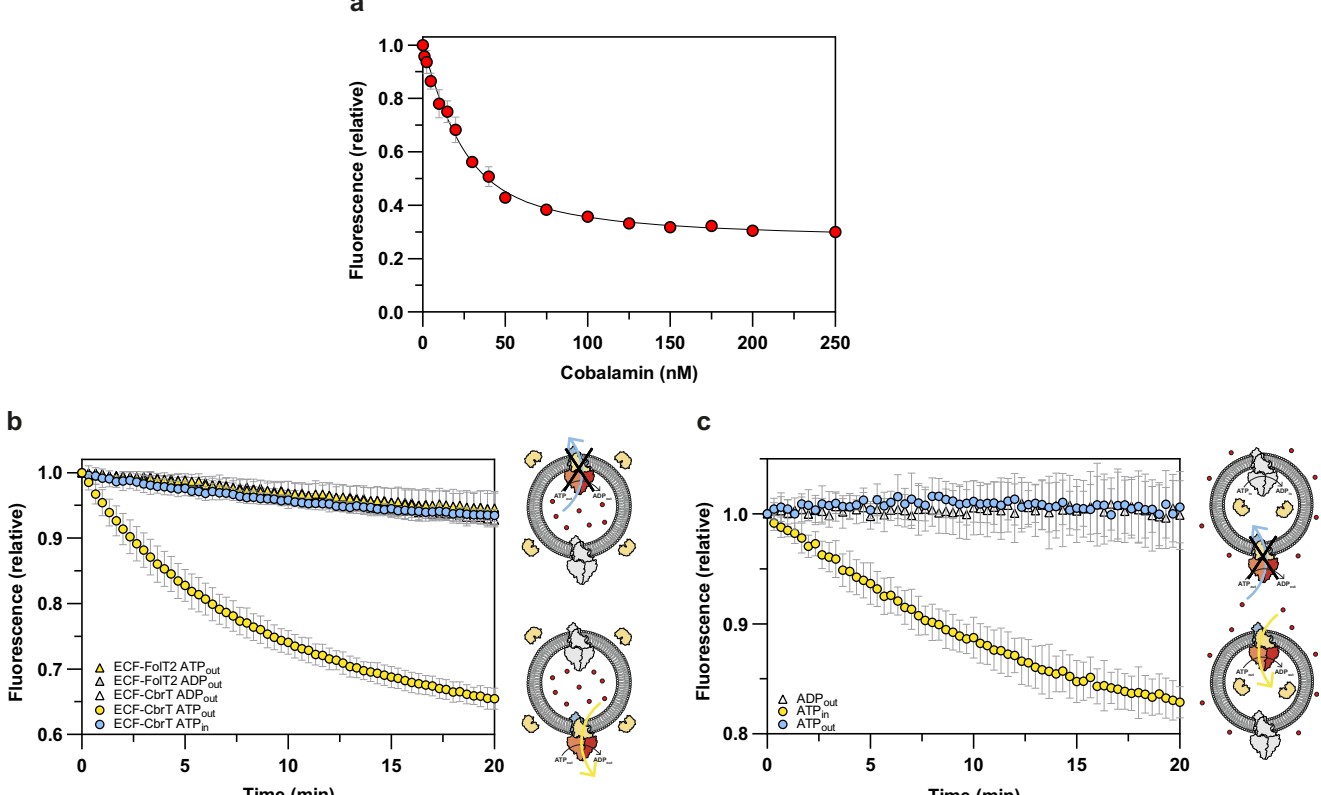

**Fig. 1 | Fluorescence-based transport assay for cobalamin. a** Fluorescence quenching of 25 nM $BtuF_{488}$ in solution in the presence of various concentrations of cobalamin. The $K_d$ value is 11.5 nM. **b** Release of cobalamin from proteoliposomes reconstituted with the cobalamin transporter ECF-CbrT or the folate transporter ECF-FolT2. Proteoliposomes were loaded with 13 μM cobalamin and 25 nM of the sensor $BtuF_{488}$ was added to the external buffer. The release of cobalamin was monitored for liposomes reconstituted with the cobalamin-specific transporter ECF-CbrT in the presence of 10 mM external Mg-ATP (yellow circles), 10 mM external Mg-ADP (white triangles) or 10 mM luminal Mg-ATP (blue circles). The folate-specific transporter ECF-FolT2, which is closely related to ECF-CbrT, was used as negative control in the presence of 10 mM external Mg-ATP (yellow triangles) or 10 mM external Mg-ADP (grey triangles). Cartoons on the right side of the graph explain the experimental setup. The liposome membrane is shown as grey circle, the sensor in the external buffer in yellow, cobalamin as red dots, the active ECF-CbrT population in colours, and the inactive population in grey. Export and import directions are indicated by blue and yellow arrows, respectively. The black cross indicates that export is not catalyzed by ECF-CbrT. **c** Accumulation of cobalamin into liposomes reconstituted with ECF-CbrT. Proteoliposomes were loaded with 2 μM $BtuF_{488}$ inside the lumen and 65 μM cobalamin was added externally. Transport activity was measured in the presence of 10 mM internal Mg-ATP (yellow circles), 10 mM external Mg-ATP (blue circles) or 10 mM external Mg-ADP (white triangles). Cartoons on the right side of the graph explain the experimental setup, as in (**b**). **a**–**c** The data were obtained from biological duplicates each containing technical triplicates. The data is presented as the mean with error bars indicating the standard deviation calculated from all individual data points ($n = 6$).

purified transporters, reconstituted in liposomes, we chose to continue with the protein from *M. marinum*, because the wild-type protein is less prone to aggregation upon purification in detergent solution than its *M. tuberculosis* counterpart. To conclusively demonstrate that the $BacA_{Mm}$ is indeed responsible for the import of cobalamin, we purified and reconstituted $BacA_{Mm}$ into liposomes (Supplementary Fig. 1d). We then assessed its ability to release cobalamin from the liposomes from the *trans*-to-the *cis* side of the protein (physiological import direction) using the fluorescence-based transport assay presented above. For experiments with $BacA_{Mm}$ we initially encapsulated a 5-fold higher concentration (65 μM) of cobalamin inside the lumen of the proteoliposomes than used with ECF-CbrT, as the BacA proteins seem to lack a high-affinity binding site for the substrate. In later experiments (see below) we used concentrations ranging from 0.5 μM to 100 μM cobalamin. Similar to ECF-CbrT, we also observed an ATP-dependent quenching of the fluorescence signal over time in the presence of externally added nucleotides for $BacA_{Mm}$ (yellow circles, Fig. 2a). We did not observe quenching of the sensor in the presence of external Mg-ADP (white triangles, Fig. 2a) or the non-hydrolysable ATP analogue AMP-PNP (grey circles, Fig. 2a), demonstrating that the transport of cobalamin is strictly coupled to the hydrolysis of ATP. This

experiment shows that a purified member of the BacA family alone is sufficient for transport of cobalamin in the physiological import direction.

It has been speculated that in addition to its import function, $BacA_{Mt}$ may also mediate the export of some antibiotics and antimicrobial peptides from the cell[14,17–23]. This suggests that the protein may use different transport directions for different substrates. We then looked into the possibility that $BacA_{Mm}$ might be able to transport a single substrate (cobalamin) not only in the import direction (*trans*-to-*cis* side), but also in the export direction (*cis*-to-*trans* side). To test this, we monitored the release of cobalamin into the external buffer in the presence of lumenal cobalamin and Mg-ATP. Strikingly, we observed a time-dependent quenching of the fluorescence signal in the presence of lumenal Mg-ATP (but not in the presence of Mg-ADP), indicating that $BacA_{Mm}$ is also able to transport cobalamin in the export direction (blue circles, Fig. 2a), albeit with a 2-3-fold lower transport rate than observed for the import direction. This is in stark contrast to the strict importer ECF-CbrT, for which export activity could not be demonstrated even at higher concentrations of lumenal cobalamin (blue circles, Fig. 1b and Supplementary Fig. 3). Extensive washing during sample preparation makes it unlikely that sufficient

**a**

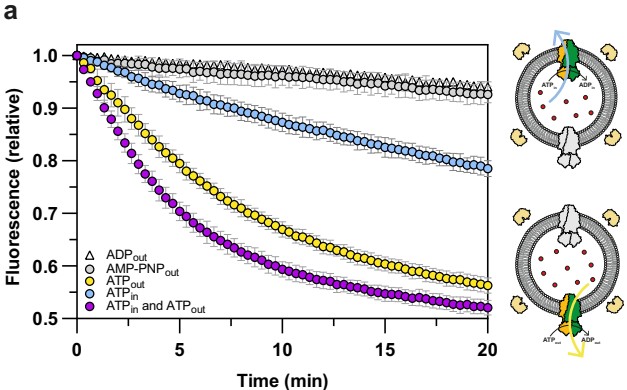

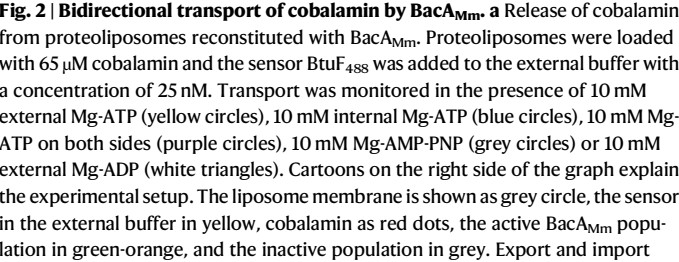

**b**

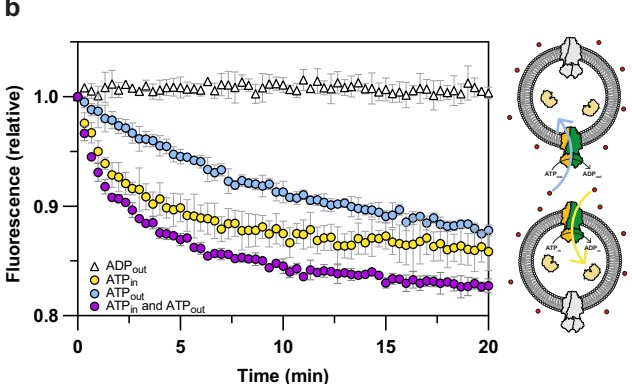

**Fig. 2 | Bidirectional transport of cobalamin by BacA$_{Mm}$. a** Release of cobalamin from proteoliposomes reconstituted with BacA$_{Mm}$. Proteoliposomes were loaded with 65 μM cobalamin and the sensor BtuF$_{488}$ was added to the external buffer with a concentration of 25 nM. Transport was monitored in the presence of 10 mM external Mg-ATP (yellow circles), 10 mM internal Mg-ATP (blue circles), 10 mM Mg-ATP on both sides (purple circles), 10 mM Mg-AMP-PNP (grey circles) or 10 mM external Mg-ADP (white triangles). Cartoons on the right side of the graph explain the experimental setup. The liposome membrane is shown as grey circle, the sensor in the external buffer in yellow, cobalamin as red dots, the active BacA$_{Mm}$ population in green-orange, and the inactive population in grey. Export and import directions are indicated by blue and yellow arrows, respectively. **b** Accumulation of cobalamin into liposomes reconstituted with BacA$_{Mm}$. Proteoliposomes were loaded with 2 μM BtuF$_{488}$ inside the lumen and 65 μM cobalamin was added externally. Transport activity was measured in the presence of 10 mM internal Mg-ATP (yellow circles), 10 mM external Mg-ATP (blue circles), 10 mM Mg-ATP on both sides (purple circles) or 10 mM internal Mg-ADP (white triangles). Cartoons on the right side of the graph explain the experimental setup, as in (**a**). **a**, **b** The data were obtained from biological duplicates each containing technical triplicates. The data is presented as the mean with error bars indicating the standard deviation calculated from all individual data points (*n* = 6).

external ATP was available for the inside-out oriented BacA population to transport cobalamin in the *trans*-to-*cis* direction. The lack of transport observed for ECF-CbrT using the same washing procedure (Fig. 1b and Supplementary Fig. 3) confirms that external ATP is sufficiently removed during sample preparation. This confirms that the release of cobalamin from the proteoliposomes resulted from actual transport activity of BacA$_{Mm}$. Finally, we prepared proteoliposomes in the presence of Mg-ATP on both sides of the membrane, loaded them with cobalamin in the lumen and monitored the flux of cobalamin to the external solution where the sensor was present. For BacA$_{Mm}$, a faster quenching of the sensor was observed compared to the quenching derived from the import or export activity alone (purple circles, Fig. 2a). This shows that both oriented protein complexes within the liposome simultaneously contribute to the release of cobalamin, which is consistent with the notion that BacA$_{Mm}$ can transport in the import and export directions. We compared this result to that of the same experiment using ECF-CbrT. Since ECF-CbrT can only transport in the import direction, we expected that the presence of ATP on both sides of the membrane would not increase the flux of cobalamin out of the liposomes, in contrast to what we observed for BacA$_{Mm}$. Indeed, there was no additive flux of cobalamin from the liposome lumen in this case, but surprisingly we found that additional lumenal Mg-ATP had the opposite effect for ECF-CbrT, as we observed a decrease in the degree of quenching of the fluorescence signal (purple circles, Supplementary Fig. 3). This effect is attributed to the re-transport of cobalamin into the proteoliposomes by import activity of the other orientation, which is possible due to the low-nanomolar binding affinity of CbrT[28]. In this case, ATP-dependent cycling of cobalamin between the compartments may lead to a steady state flux in which the level of sensor quenching is reduced, compared to unidirectional efflux only. Altogether, we show that a wild-type ABC transport system drives bidirectional transport of a single substrate.

The apparent transport rates suggest that BacA$_{Mm}$ transports cobalamin 2-3-fold more efficiently in the import direction than the export direction, but this conclusion only holds if the populations of BacA$_{Mm}$ reconstituted in the right-side-out and inside-out orientations are equal. Any deviation from the 1:1 orientation will affect the observed rates of transport. In addition, the membrane curvature could also affect the proteins oriented for transport in the import and

export directions differently. To test the effect of the protein orientation on the observed activity, we reversed the transport setup (similar to what we had done for ECF-CbrT, Fig. 1c) and examined the transport activity of BacA$_{Mm}$ by monitoring the accumulation of cobalamin into the lumen of proteoliposomes. We encapsulated the sensor within the lumen of the proteoliposomes, added cobalamin to the external buffer, and had Mg-ATP present either in the lumen, or in the external solution, or on both sides. Quenching of the sensor was observed both when Mg-ATP was co-encapsulated in the lumen of the liposomes (yellow circles, Fig. 2b), and when Mg-ATP was added to the external buffer (blue circles, Fig. 2b) confirming that BacA$_{Mm}$ transports cobalamin both in the import and export direction. Again, a 2-3-fold higher transport rate was found in the import direction compared to the export direction, indicating that transport in the *trans*-to-*cis* (import) direction is somewhat more efficient than in the *trans*-to-*cis* (export) direction, and that the transporter had been reconstituted in equal proportions in the right-side-out and inside-out orientation. When Mg-ATP was added to both the internal and external solutions, there was again an additive effect on the transport rate (purple circles, Fig. 2b).

## Transport kinetics of cobalamin in the export and import direction

To gain more insight into the unusual bidirectionality of BacA$_{Mm}$, we assessed the transport kinetics of cobalamin in both transport directions. We monitored the import of cobalamin into proteoliposomes and triggered the transport activity in the presence of increasing concentrations of cobalamin in the external buffer. A fixed concentration of Mg-ATP was present either in the lumen (*trans*-to-*cis* or import reaction) or in the external buffer (*cis*-to-*trans* or export reaction). The initial transport rates depended on the substrate concentrations for both transport directions (Fig. 3a). Remarkably, the apparent $K_M$ for cobalamin for transport in the import direction of 3.9 μM was lower than the apparent $K_M$ for transport in the export direction (40 μM, but it is noteworthy that this value may be less accurate than the one for the import direction, because we could not saturate the transport rates). The absolute $V_{max}$ values are difficult to determine from these experiments, because only a fraction of the sensors encapsulated within the liposomes is accessible to transported

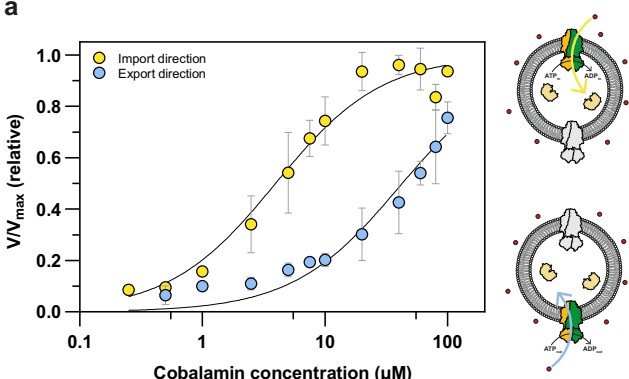

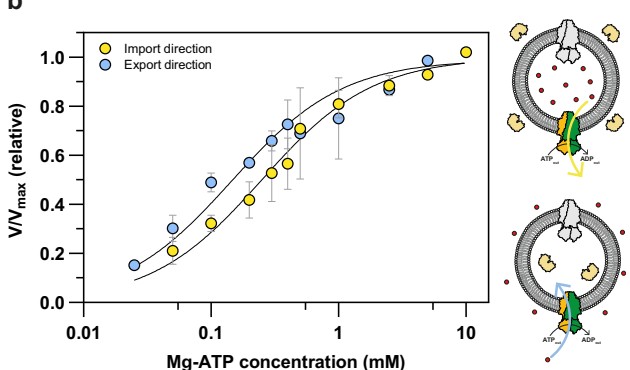

**Fig. 3 | Transport kinetics of BacA_Mm for transport of cobalamin. a** Initial cobalamin transport rates determined for BacA_Mm reconstituted into liposomes in the presence of various concentrations of cobalamin. The transport rates were tested in the presence of 10 mM Mg-ATP for the import direction (yellow) and the export direction (blue). Cartoons on the right side of the graph explain the experimental setup. The liposome membrane is shown as grey circle, the sensor in the lumen in yellow, cobalamin as red dots, the active BacA_Mm population in green-orange, and the inactive population in grey. Export and import directions are indicated by blue and yellow arrows, respectively. The $K_M$ values for cobalamin are 3.9 μM and 40 μM for transport in the import and export direction, respectively. **b** Initial transport activity of BacA_Mm reconstituted into liposomes in the presence of various concentrations of Mg-ATP. The transport rates were tested in the presence of 65 μM cobalamin for the import direction (yellow) and in the presence of 100 μM cobalamin for the export direction (blue). The $K_M$ values for Mg-ATP are 0.25 mM and 0.15 mM for transport in the import and export direction, respectively. Cartoons on the right side of the graph explain the experimental setup, as in (**a**). **a**, **b** The data were obtained from biological duplicates each containing technical triplicates. Linear regression was used to determine the initial transport rates of the averaged independent experiments. The activity was then normalised to the highest activity within the data set. The mean of the two averaged biological duplicates was plotted with error bars indicating the standard deviation calculated between the two biological repeats ($n = 2$). The apparent $K_M$ and $V_{max}$ values were derived using the Michaelis-Menten equation.

cobalamin due to the expected heterogeneity in the proteoliposome population (including multilamellar liposomes, and liposomes containing sensors without reconstituted active transporters). Nonetheless, it is possible to compare the relative differences in $V_{max}$ between import and export activity. Although there is some variation between experiments, this comparison shows that the $V_{max}$ values in the two transport directions are very similar and differ by less than two-fold. Altogether, this data demonstrates that BacA_Mm operates somewhat more efficiently as a cobalamin importer at physiological concentrations, which is consistent with the observation that the protein mediates the transport of cobalamin in the import direction in vivo (Supplementary Fig. 2). Finally, we tested the transport dependency on the Mg-ATP concentration for both transport directions, and found that the apparent $K_M$ for ATP was nearly identical for both the import and export activity (Fig. 3b).

## Discussion

Conventionally, assays to detect transport across liposomal membranes make use of radioactive substrates. While these assays have proven to be sensitive, robust and capable of accurate quantification, the throughput is often very low. In addition, radioactive substrates are increasingly expensive, generate radioactive waste, and their use is subject to stringent regulation. In the particular case of the substrate cobalamin, the use of $^{57}$Co-cyanocobalamin requires elaborate safety precautions[28]. Hence, development of alternative transport assays that support high-throughput experiments are desirable.

Methods established in the past to detect cobalamin in solution or in cellular or liposomal compartments without the use of radioactive substrates[32–41] often do not allow for high-throughput measurements, or are incompatible with assays to measure transport in both import and export direction. We chose to use a protein-based sensor derived from the SBP BtuF[25]. This sensor exploits the fluorescence quenching of a conjugated fluorophore by cobalamin upon binding of the substrate to the protein[25,42] and has been used to monitor the presence of cobalamin in liposomes at the single-molecule level[25]. Similar sensors have been used to measure binding of cobalamin to BtuF in bulk solution[26,43] and at single molecule level[44,45]. The sensor is membrane

impermeable, binds cobalamin with low nanomolar $K_d$, and allows for sensitive detection of substrate imported into, or exported from the liposomal lumen (Supplementary Fig. 4). While the high-affinity binding property of BtuF_488 on one hand provides sensitivity to the transport assay, it may on the other hand also act as a sink, making it difficult to demonstrate transport of cobalamin against its concentration gradient. We established a fluorescence-based transport assay using the well-characterized cobalamin transporter ECF-CbrT, and then used it to characterize the mycobacterial ABC transporter BacA.

Our study reveals that the mycobacterial type IV ABC transporter BacA_Mm supports the ATP-driven transport of cobalamin in both the import and export direction, which is exceptional among ABC transporters, as these proteins generally use the free energy from the hydrolysis of ATP to ensure unidirectionality[1–7]. Physiologically relevant bidirectional substrate movement in the ABC transporter family has only been observed in CFTR[8], which is mechanistically distinct from all ABC transporters and acts as an ATP-gated channel rather than a transporter. It is noteworthy that a few other type IV ABC transporters have been genetically engineered to change their transport mode by introducing different type of mutations. For instance, mutations in the D-loop and the outer gate of the human ABC transporters TAP[46] and ABCB1[47], respectively, converted the exporters into facilitators that translocate their substrates along the concentration gradient. In contrast to BacA_Mm, facilitated diffusion of peptides by the mutated TAP transporter was no longer coupled to the hydrolysis of ATP. In the case of the multidrug efflux pumps ABCB1[48] and ErfCD[49], mutations in the substrate binding site have been demonstrated to reverse the direction of transport for some of their substrates, while the transport directionality is maintained for others. However, to the best of our knowledge, bidirectional transport has not been presviouly observed for a native ABC transporter. It is possible that bidirectionality may be found in more ABC transporters, but has so far remained undetected because the assay conditions prevented detection.

The unusual bidirectionality is consistent with previous studies in whole cells showing that on the one hand BacA_Mt facilitates the import of several hydrophilic compounds[9,11,13–16], including cobalamin, whilst

on the other hand there are also indications that the protein may be involved in the export of antimicrobial compounds[14,17–23]. From an evolutionary perspective, BacA may be a generalist transport system with the ability to transport a wide range of hydrophilic substrates not necessarily related to physiologically relevant processes, which over time may evolve into a specialized transporter[50]. Although $BacA_{Mm}$ transports cobalamin in both directions, the apparent $K_M$ for cobalamin is approximately 10-fold lower for transport in the import direction (Fig. 3a), while $V_{max}$ values are similar. Therefore, in the presence of physiological concentrations of cobalamin, operation of $BacA_{Mm}$ could lead to a low level of accumulation, albeit much less what the hydrolysis of ATP could achieve if all free energy was used for accumulation. The capacity to export the micronutrient may be of physiological relevance, because *M. marinum* (in contrast to *M. tuberculosis*) is a bacterium that can synthesise cobalamin[51,52]. Cobalamin is often shared between cobalamin producers and consumers within microbial communities[53,54]. While multiple membrane transport systems have been identified that are used to import cobalamin[55], it has remained elusive how synthesised cobalamin is released into the environment[54]. It is likely that cobalamin producing microbes live in symbiosis with cobalamin auxotrophic organisms and therefore share the vitamin via the export through membrane transporters[54,56]. Other B-type vitamins are hypothesised to be shared through the active export via secondary active bidirectional transporters[54]. In contrast, BacA is a primary active ATP-driven transporter. We speculate that transporters belonging to the BacA family[57] may contribute to the release of cobalamin, and possibly other shared metabolites, into the surroundings.

It remains elusive how, unlike other well-characterised ABC transporters, $BacA_{Mm}$ can mechanistically transport substrates in both directions. The structural information of the BacA family is scarce, as only a single structure of $BacA_{Mt}$ has been solved, reflecting the nucleotide-bound occluded state with an unidentified substrate trapped in its cavity[9]. The protein possesses a large cavity of ~7700 $Å^3$ lined with polar and negative residues, which can explain its ability to transport multiple structurally unrelated hydrophilic compounds. However, structures under turnover conditions may be needed to gain insights into the bidirectional transport mechanism of $BacA_{Mm}$.

## Methods

### Cloning
$bacA_{Mm}$ was amplified from the genomic DNA of *M. marinum* and cloned into the *E. coli* expression vector pBXC3H from the FX cloning kit[58]. Site-directed mutagenesis was performed using partially overlapping primers to remove all cysteines from ECF-CbrT, as we noted that the wild-type protein had the tendency to be purified in the absence of the S-component CbrT. All primers are listed in Supplementary Table 2.

### Overexpression and membrane vesicle preparation
The membrane proteins ECF-CbrT[28], ECF-FolT[59] and $BacA_{Mm}$[9] were expressed as described previously. Briefly, *E. coli* MC1061 cells were grown in 5 L baffled flasks filled with 2 L LB miller broth containing 100 μg ml$^{-1}$ ampicillin at 37 °C while continuously shaking. Expression of the protein was induced at an $OD_{600}$ of 0.8 with 0.01% (w/v) and 0.02% (w/v) L-arabinose (Sigma Aldrich) for both ECF transporters and $BacA_{Mm}$, respectively. Cells were harvested 3 h after induction by centrifugation (20 min, 6000 x g, 4 °C) and were washed with ice-cold 50 mM potassium phosphate (KPi), pH 7.5. Cells were resuspended in breaking buffer (50 mM KPi, pH 7.5, 10% (v/v) glycerol, 2 mM MgSO₄, 0.5 mM PMSF (Roth) and 4 μg mL$^{-1}$ DNase I (Sigma-Aldrich)) and were lysed by three passages through a high-pressure homogeniser HPL6 system (Maximator) at 25 kpsi. Cell debris was removed by high-speed centrifugation (40 min, 31,000 x g, 4 °C), after which the crude membranes were collected by ultracentrifugation (150 min, 186,000, 4 °C). Membranes were homogenised in storage buffer (50 mM KPi pH

7.5, 10% (v/v) glycerol) and stored in aliquots at −70 °C until further use after flash freezing in liquid nitrogen. The total protein concentration was determined using the bicinchoninic acid assay kit (Thermo Fisher Scientific). For the preparation of membrane vesicles containing $BacA_{Mm}$, the breaking buffer and storage buffer was supplemented with 1 mM DTT, and a complete EDTA-free protease inhibitor tablet (Roche) was added to the breaking buffer.

### Expression, purification and labelling of BtuF
Expression of BtuF and cell lysis was accomplished with the same procedure as described above with minor modifications. The temperature during cell growth was reduced to 25 °C when the cells reached an $OD_{600}$ of 0.5 and protein was expressed at an $OD_{600}$ of 0.8 with 0.01% (w/v) L-arabinose. Cell lysis was done in the presence of 1 mM DTT, and the supernatant obtained after removal of the cell debris was directly used for subsequent purification steps. The supernatant was incubated for 1 h with 2 mL bed volume of Nickel-Sepharose 6 Fast flow beads (Cytiva) that was pre-equilibrated with 20 column volumes (CV) of 50 mM KPi, pH 7.5, 300 mM NaCl, 10% (v/v) glycerol and 1 mM DTT. The resin was then packed into an Econo-Pac column (BioRad) and the unbound proteins were allowed to flow through. Non-specifically bound proteins were removed by 40 column volume (CV) of wash buffer (50 mM KPi, pH 7.5, 300 mM NaCl, 50 mM imidazole, pH 7.5, 10% (v/v) glycerol, 1 mM DTT). BtuF was eventually eluted from the column with elution buffer (50 mM KPi, pH 7.5, 300 mM NaCl, 350 mM imidazole, pH 7.5, 10% (v/v) glycerol, 1 mM DTT). Fractions containing most protein were further purified by size-exclusion chromatography (SEC) using a Superdex 200 increase 10/300 GL column (Cytiva) equilibrated with SEC buffer (50 mM KPi, pH 7.5, 200 mM NaCl, 10% (v/v) glycerol, 1 mM DTT). 21 nmol BtuF was rebound to 0.3 mL bed volume of Nickel-Sepharose 6 Fast flow beads, that was pre-equilibrated with 20 CV SEC buffer. DTT and glycerol were removed with 9 CV labelling buffer (50 mM KPi, pH 7.5, 200 mM NaCl), after which the protein was immediately incubated at 4 °C for 2 h with a 2-fold molar excess of AF488 C5 maleimide (Jena Science) or XDF-488 C5 maleimide (AAT Bioquest) dissolved in labelling buffer. Excess fluorophore was removed with 20 CV washing buffer followed by the elution of the labelled protein (50 mM KPi, 7.5, 200 mM NaCl, 350 mM imidazole, pH 7.5, 5% (v/v) glycerol). Labelled protein was further purified by SEC using a Superdex 200 increase 10/300 GL column equilibrated with SEC buffer (50 mM KPi, pH 7.5, 150 mM NaCl, 5% (v/v) glycerol). Aliquots were flash frozen in liquid nitrogen and stored at −70 °C until further use.

### Binding of cobalamin to BtuF₄₈₈
25 nM of BtuF₄₈₈ in 50 mM KPi, pH 7.5 and 150 mM NaCl solution was mixed with various concentrations of Cyano-cobalamin (Acros Organics), ranging from 1 nM to 3 μM, to determine the dissociation constant. The mixture was incubated for 5 min at 30 °C in a black polystyrene 96-well plate (Greiner) coated with bovine serum albumin (BSA, Roth), to reduce interactions of BtuF₄₈₈ with the 96-well plate. The fluorescence intensity was then measured on a Spark 10 M microplate reader (TECAN) using an excitation wavelength of 485 nm (bandwidth 5) and an emission wavelength of 520 nm (bandwidth 10). The normalised decrease in fluorescence was plotted against the substrate concentration. The measurements were performed as biological duplicates each containing technical triplicates. The $K_D$ was derived by fitting the formula $Y = 1 - \left( \frac{A\left((Et+X+Kd) - \sqrt{(Et+X+Kd)^2 - 4*Et*X}\right)}{2} \right)$ in GraphPad Prism (version 10.0.2).

### Purification of ECF-CbrT and ECF-FolT2
Membrane vesicles containing 40–60 mg of total protein were solubilized in solubilisation buffer (50 mM KPi, pH 7.5, 300 mM NaCl, 10%

(v/v) glycerol and 1% (w/v) dodecyl-β-maltoside (DDM, Glycon)) for 1 h while gently agitating. Insoluble material was subsequently removed by ultracentrifugation (25 min, 444,000 x g, at 4 °C). The supernatant was incubated for 1 h with 0.5 mL bed volume of Nickel-Sepharose 6 Fast flow beads that was pre-equilibrated with 20 CV of wash buffer (50 mM KPi, pH 7.5, 300 mM NaCl, 0.05% (w/v) DDM, 50 mM imidazole, pH 7.5) while gently agitating. The resin was then packed into a polypropylene column (BioRad) and the unbound proteins were allowed to flow through. Non-specifically bound protein was removed with 30 CV wash buffer. Bound proteins were eluted with elution buffer (50 mM KPi, pH 7.5, 300 mM NaCl, 0.05% (w/v) DDM and 300 mM imidazole, pH 7.5) in fractions of 450 μl–800 μl–800 μl. The elution fraction containing most protein was further purified by SEC using a Superdex 200 increase 10/300 GL column equilibrated with SEC buffer (50 mM KPi, pH 7.5, 200 mM NaCl, 0.05% (w/v) DDM). The peak fractions corresponding to the purified protein were pooled and used for reconstitutions into liposomes.

### Purification BacA$_{Mm}$

Purification was performed similar to ECF-CbrT and ECF-FolT2 with minor modifications. 50 mM KPi, pH 7.5 was replaced for Tris, pH 8.0 and the concentration was lowered to a concentration of 25 mM except for the solubilisation. DDM was replaced for DDM/CHS (10:1) to stabilise the protein. The elution buffer was supplemented with 350 mM imidazole, pH 7.5. All buffers were supplemented with 1 mM DTT.

### Reconstitution into liposomes

Freshly purified ECF-CbrT, ECF-FolT2 and BacA$_{Mm}$ were reconstituted into Triton X-100 destabilised liposomes consisting of *E. coli* polar lipid extract (Avanti Polar Lipids) and egg phosphatidylcholine (Avanti Polar Lipids) at a 3:1 ratio dissolved in 50 mM KPi, pH 7.5, as described previously[60]. Briefly, *E. coli* polar lipid extract, prepared from *E. coli* total lipid extract (Avanti Polar Lipids) and egg phosphatidylcholine (Avanti Polar Lipids) dissolved in chloroform were mixed at a 3:1 ratio (w/w) and dried in a rotavopor. The obtained lipid film was dissolved to a concentration of 20 mg ml$^{-1}$ in 50 mM KPi, pH 7.5 buffer, and then extruded eleven times through a 400 nm polycarbonate filter (Avestin). The liposomes were destabilised by adding Triton X-100, until half of the initial absorbance at 540 nm was reached. ECF transporters and BacA$_{Mm}$ were added with a protein: lipid ratio of 1:400 (w/w) and 1:250 (w/w), respectively. The mixture was then incubated at 4 °C for 30 min while gently rotating, after which the detergent was removed by the addition of BioBeads SM-2 (Bio-Rad) in four steps. Proteoliposomes were collected by ultracentrifugation (30 min, 444,000 x g, at 4 °C), resuspended in 50 mM KPi, pH 7.5 to a concentration of 40 mg ml$^{-1}$ lipids, flash-frozen in aliquots and stored in liquid nitrogen until further use. All buffers were supplemented with 1 mM DTT when liposomes were reconstituted with BacA$_{Mm}$.

### Fluorescence-based release assay for cobalamin

Proteoliposomes dissolved in assay buffer (50 mM KPi, pH 7.5 and 150 mM NaCl) were loaded with either 13 μM or 65 μM Cyanocobalamin and optionally 10 mM Mg-ATP (Roche) through five freeze-thaw cycles, followed by extrusion of eleven passes through a 400 nm polycarbonate filter (Cytiva). Excess of cobalamin was removed by ultracentrifugation (30 min, 444,000 x g, at 4 °C) and the proteoliposomes were once more washed in assay buffer. Transport reactions were performed in black polystyrene 96-well plate (Greiner) using proteoliposomes resuspended in assay buffer with a lipid concentration of 2.5 mg ml$^{-1}$ in a final volume of 200 μl. The sensor BtuF$_{488}$ was supplemented to the external buffer with a concentration of 25 nM. Mg-ATP or Mg-ADP (Sigma Aldrich) was added to a final concentration of 10 mM (unless stated otherwise) to trigger transport in the import direction. The transport activity was immediately monitored on a Spark 10 M microplate reader (TECAN) by measuring the fluorescence signal every 20 s at an excitation wavelength of 485 nm (bandwidth 5) and an emission wavelength of 520 nm (bandwidth 10) using the instruments' SparksControl software version 2.3 (TECAN). All steps were performed in the presence of 1 mM DTT for liposomes reconstituted with BacA$_{Mm}$. The measurements were performed at 30 °C as biological duplicates each containing technical triplicates. Signals were normalised to the first reading after the addition of nucleotides.

### Fluorescence-based accumulation assay for cobalamin

The fluorescence-based accumulation assay was performed as described for the fluorescence-based transport assay with minor modifications. Proteoliposomes were loaded with 2 μM BtuF$_{488}$ and optionally 10 mM Mg-ATP or Mg-ADP through two freeze-thaw cycles, followed by twenty-three passes through a 400 nm polycarbonate filter. Transport reactions were performed using proteoliposomes with a lipid concentration of 1.25 mg ml$^{-1}$ or 2.5 mg ml$^{-1}$. Mg-ATP or Mg-ADP was optionally added to the external buffer to a final concentration of 10 mM to trigger transport in the export direction. Cyano-cobalamin was added to a final concentration of 65 μM (unless stated otherwise) to trigger transport.

### Cobalamin-dependent growth assays

Growth assays were performed as previously described[31]. Briefly, the cobalamin-deficient *E. coli* ΔFEC strain was transformed with various expression vectors to express BacA$_{Mt}$, ECF-CbrT or ECF-FolT2. Single colonies were then innoculated into M9 medium supplemented with 0.00001% (w/v) arabinose (Sigma-Aldrich) and 50 μg ml$^{-1}$ methionine (Sigma-Aldrich). Cells were grown at 37 °C for 24 h and were then diluted with a ratio of 1:250 into 200 μl fresh M9 medium supplemented with 0.00001% (w/w) arabinose and either 1 nM cyanocobalamin or 50 μg ml$^{-1}$ methionine (Sigma-Aldrich) in 96-well plates (Greiner). Plates were sealed with sterile and gas-permeable foil (Breathe-Easy, Diversified Biotech) and incubated at 37 °C for 24 h. The OD$_{600}$ was measured every 10 min on a SpectraMax ABS plus Microplate reader (Molecular Devices). The measurements were performed as biological triplicates each containing technical triplicates and collected using the instruments' SoftMax Pro version 7.1.2 (Molecular Devices).

### Reporting summary

Further information on research design is available in the Nature Portfolio Reporting Summary linked to this article.

## Data availability

The data that support this study are available from the corresponding authors upon request. The previously solved structure of BacA$_{Mt}$ in the AMP-PNP bound state is available through the Protein Data Bank (PDB) under the accession code 6TQF. Protein sequences used in this study are available through UniProt under the accession codes P37028 for BtuF, B2HSW5 for BacAMm, Q1GBJ0 for EcfA, Q1GBI9 for EcfA', Q1GBI8 for EcfT, Q1G292 for FolT2, and Q1G7W0 for CbrT. The source data underlying Figs. 1a–c, 2a,b, 3a,b and Supplementary Fig. 2, Supplementary Fig. 3 and Supplementary Fig. 6 are provided as Source Data file. Source data are provided with this paper.

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

## Acknowledgements

We would like to thank Wilbert Bitter and Alexander Speer (Vrije Universiteit Amsterdam) for kindly providing us the genomic DNA of *Mycobacterium marinum*. D.J.S. acknowledges funding from the Dutch Research Council: NWO TOP Grant 714.018.003. S.N.L. acknowledges funding from the FEBS long term fellowship. As part of the COFUND project oLife, C.T. acknowledges funding from the European Union's Horizon 2020 research and innovation programme under the Grant Agreement 847675.

## Author contributions

M.N., C.T. and D.J.S. designed the research. M.N. performed protein expression, purifications and fluorescence-based transport assays. M.N. and S.N.L. performed cloning. S.N.L. prepared lipids for proteoliposomes preparation. C.T. performed protein expression for ECF-FolT2. M.N. and D.J.S. prepared the manuscript with input from all authors.

## Competing interests

The authors declare no competing interests.
