## [Peer Review File · Nature Communications]

Bidirectional ATP-driven transport of cobalamin by the mycobacterial ABC transporter BacAReviewer #1 (Remarks to the Author):

It was with great pleasure I read this interesting article by Marc Nijland, Dirk Slotboom and colleagues. In the manuscript, the authors studied cobalamin (Vitamin B12) transport in proteoliposomes using reconstituted ABC transporters. VitB12 detection was performed using a fluorescently labelled version of BtuF, a periplasmic VitB12 binding protein from *E. coli*, whose fluorescence becomes quenched when VitB12 binds. Next to the advantage that the assay can be performed without the use of radioactive material, modified BtuF does not cross the lipid bilayer and thus can reliably report on the VitB12 concentration inside the liposome lumen or in the bulk solution. In a first set of experiments, the authors reconstituted the ECF-type ABC importer (Type III system) called ECF-CbrT and showed that it acts as a strict importer without any discernible export activity. This experimental outcome was expected; but importantly, it set the stage for a very remarkable observation. Namely, when the authors reconstituted BacA (*M. marinum* homologue), which is Type IV ABC transporter previously shown to import VitB12 in *M. tuberculosis*, they could show that this transporter is capable of VitB12 import as well as VitB12 export, depending on which side of the membrane VitB12 is added. Transport only occurs when ATP is present, but not when ADP is supplied. This unique finding adds an unprecedented and interesting layer on the functional diversity of type IV ABC transporters. Generally, ABC transporters couple the energy gained from ATP hydrolysis to the uphill transport of substrates. In the case of classical importers (Type I – III ABC transporters), it is strictly the import direction (as confirmed again in this paper for ECF-CbrT), whereas for the classical exporters (Type IV and V ABC transporters), it is usually the export direction, with notable exceptions such as for example BacA recently shown to import VitB12.

In this paper, the authors show now in a convincing dataset involving proteoliposomes transport assays of very high technical quality, that BacA can in fact transport VitB12 in both directions (and intriguingly with different apparent affinities of transport), thereby deviating from the “dogma” that ABC transporters operate per definition unidirectionally.

The paper is well written and the authors did a great job in explaining the *in vitro* transport assays in a simple manner that is easy to follow. The presented results are of exceptional technical quality (i.e. high transport activities that can be interpreted with high confidence) when compared to other such studies in the ABC transporter field. I also appreciated to read the discussion, which puts the data into a larger evolutionary context; i.e. VitB12 export likely plays an important physiological role in the case of saprophytic mycobacteria which in fact produce Vit12, as they need the export system to share this precious molecule with other bacteria in the context of microbial communities. Further, the authors embed and interpret their main finding eloquently into the larger context of ABC transporters and transport proteins in general.

I thus consider this study as a very important scientific contribution to the ABC transporter field and beyond.

Nevertheless, I listed a few points below, which may be of interest/help for the authors to clarify a few technical points.

Major points

- 1) The authors state in the abstract that “BacA is mechanistically distinct from other members of the ABC transport family and facilitates ATP-driven facilitated diffusion”. However, in the results section, the authors seem to be less explicit in interpreting their data in this regard. While their interpretation of BacA being a ATP-driven facilitator of VitB12 conceptually makes perfect sense and can explain how this ABC transporter can mediate VitB12 transport downhill its concentration gradient in either direction, I still would wonder whether the authors can exclude the possibility that in one of the transport directions (e.g. the import direction?), BacA mediates uphill transport.
- 2) Another potential technical caveat is the high affinity of the VitB12 sensor BtuF. I am sure, the authors considered the fact that BtuF acts as a sink, thus driving VitB12 transport into the direction of the sensor. Given the authors look here at facilitated transport (and do not claim uphill transport to happen), I do not think this is really problematic for the interpretation of the results. But nevertheless, a short note on this point in the paper might not harm.
- 3) The authors did not include any data on BacA that carries mutations to abrogate ATP hydrolysis.

While I appreciate the fact that the authors have conducted the transport assays using ADP as control (and thereby established that ATP is needed for VitB12 transport mediated by BacA), it would nevertheless have been interesting to see how a Walker B EtoQ mutant would have behaved in terms of transport.

Minor points

1) Line 140: I was surprised to see such a strong quenching in the setup wherein VitB12 was entrapped in the (tiny) liposome lumen and transported into the bulk solution where BtuF was present, which has a much larger volume. Did the author calculate/determine the maximal quenching of BtuF that can be achieved in this experimental setup?

2) Line 140: The legend says "The data were obtained from biological duplicates containing each technical triplicates. The data is presented as the mean with error bars indicating the standard deviation." What remains unclear (here and also in further instances) how the standard deviation was calculated: did the authors simply take the 6 data points to compute mean and STDEV, or did they calculated the STDEVs for the two biological replicates separately and performed an error propagation calculation to arrive at the final STEDEV?

3) Line 210: Author statement: "Again, a 2-3-fold higher transport rate was found in the import direction compared to the export direction, indicating that transport in the trans-to-cis (import) direction is somewhat more efficient than in the trans-to-cis (export) direction, and that the transporter had been reconstituted in equal proportions in the right-side-out and inside-out orientation." It would have been nice/desirable, if the authors would have estimated the orientation of insertion of the BacA using an orthogonal method, e.g. tag-cleavage by a protease followed by Western blotting.

4) Line 218: "10 mM externa Mg-ATP (blue circles),..." -> typo: external

Reviewer #2 (Remarks to the Author):

The study by Nijland et al. characterizes the direction of transport of the ABC transporter BacA. According to the current paradigm, ABC transporters are unidirectional, i.e. they can either import or export substrates, but never transport in more than one direction. This distinguishes ABC transporters from secondary transporters, which are bidirectional by definition. Here, the authors propose that BacA is indeed an ABC transporter that allows transport in two directions. This is an extraordinary claim, and the authors are aware that it requires extraordinary evidence.

The experimental design and its execution are very strong and result in convincing and complete data sets. For example, my concern that the reduced transport rates (Fig. 2, BacA transport) could be explained by a reduced internal concentration of ATP and/or cobalamin due to ATP hydrolysis or cobalamin export during sample preparation was properly addressed by monitoring transport in the opposite direction as well.

Therefore, I can only share the authors' conviction that they have measured bidirectional transport of the ABC transporter BacA, which is a sensational finding.

Major comment

Line 133, 'co-encapsulated'. The ability to export cobalamin from cis to trans depends on the presence of ATP. Clearly, ATP is co-encapsulated. However, 1) several ABC transporters show uncoupled ATP hydrolysis; and 2) ATP hydrolysis is even more likely to occur in the presence of a substrate. If this is the case for ECF-CbrT, this could explain the lack of transport in the export direction: either no ATP is left or all substrate has already left the liposome due to cis-to-trans export activity during sample preparation. Please explain why this scenario is not considered.

Minor comments

The authors have specifically designed their assay to determine the direction of transport. Please comment on the ability of more conventional transport experiments (e.g. those using radioactive substrate) to reveal (or hide) bidirectional transport. What are the chances that this is a more common feature that has gone undetected due to a bias in the experimental system? In this regard, please explain why ECF-CBrT (line 111) can be considered "strictly unidirectional" based on ref28 and 29.

Line 172, "transport [...] export direction". Needless to say, any remaining external ATP from the co-encapsulation procedure would also have led to the observation of cobalamin leaving the proteoliposomes. However, the author followed a procedure of washing and resuspension that makes it unlikely that any external ATP is present. The authors should emphasize this in their text.

Line 229, the affinity is clearly different. I am concerned about the three digit precision for transport in the export direction, which seems unrealistically high. This value assumes that transport is saturated, but it is not clear from fig3a that this is indeed the case for the export direction. Instead, I suggest that the authors rephrase to indicate that the K_m -app is at most 38 μ M.

Line 234, "less than twofold" -> what data are the authors referring to? Is this panel 3A? If so, please confirm that the same V_{max} value was used for normalization.

Fig. 3b, please indicate at what cobalamin concentration these measurements were made.

Line 60, "smuggles" suggests some degree of secrecy. An alternative would be to indicate that toxic compounds are taken up as part of a moonlighting activity of the protein.

Line 168: import should be trans-to-cis

Line 219: externa -> external

Line 230: "only a fraction" -> please explain this statement. It is not clear why the authors assume that only a fraction of the sensors are accessible. Is this due to the expected heterogeneity in the distribution of the sensors across the liposomes? Or has this been demonstrated (where?)?

Line 245: The first paragraph does not seem to be relevant to this manuscript and I suggest deleting it.

We thank all reviewers for their constructive comments, and are delighted that they are as enthusiastic about the work as we are.

[Reviewer #1 Remarks to the author]

It was with great pleasure I read this interesting article by Marc Nijland, Dirk Slotboom and colleagues. In the manuscript, the authors studied cobalamin (Vitamin B12) transport in proteoliposomes using reconstituted ABC transporters. VitB12 detection was performed using a fluorescently labelled version of BtuF, a periplasmic VitB12 binding protein from *E. coli*, whose fluorescence becomes quenched when VitB12 binds. Next to the advantage that the assay can be performed without the use of radioactive material, modified BtuF does not cross the lipid bilayer and thus can reliably report on the VitB12 concentration inside the liposome lumen or in the bulk solution.

In a first set of experiments, the authors reconstituted the ECF-type ABC importer (Type III system) called ECF-CbrT and showed that it acts as a strict importer without any discernible export activity. This experimental outcome was expected; but importantly, it set the stage for a very remarkable observation. Namely, when the authors reconstituted BacA (*M. marinum* homologue), which is Type IV ABC transporter previously shown to import VitB12 in *M. tuberculosis*, they could show that this transporter is capable of VitB12 import as well as VitB12 export, depending on which side of the membrane VitB12 is added. Transport only occurs when ATP is present, but not when ADP is supplied.

This unique finding adds an unprecedented and interesting layer on the functional diversity of type IV ABC transporters. Generally, ABC transporters couple the energy gained from ATP hydrolysis to the uphill transport of substrates. In the case of classical importers (Type I – III ABC transporters), it is strictly the import direction (as confirmed again in this paper for ECF-CbrT), whereas for the classical exporters (Type IV and V ABC transporters), it is usually the export direction, with notable exceptions such as for example BacA recently shown to import VitB12.

In this paper, the authors show now in a convincing dataset involving proteoliposomes transport assays of very high technical quality, that BacA can in fact transport VitB12 in both directions (and intriguingly with different apparent affinities of transport), thereby deviating from the “dogma” that ABC transporters operate per definition unidirectionally.

The paper is well written and the authors did a great job in explaining the in vitro transport assays in a simple manner that is easy to follow. The presented results are of exceptional technical quality (i.e. high transport activities that can be interpreted with high confidence) when compared to other such studies in the ABC transporter field. I also appreciated to read the discussion, which puts the data into a larger evolutionary context; i.e. VitB12 export likely plays an important physiological role in the case of saprophytic mycobacteria which in fact produce VitB12, as they need the export system to share this precious molecule with other bacteria in the context of microbial communities. Further, the authors embed and interpret their main finding eloquently into the larger context of ABC transporters and transport proteins in general.

I thus consider this study as a very important scientific contribution to the ABC transporter field and beyond. Nevertheless, I listed a few points below, which may be of interest/help for the authors to clarify a few technical points.

- 1) The authors state in the abstract that “BacA is mechanistically distinct from other members of the ABC transport family and facilitates ATP-driven facilitated diffusion”. However, in the results section, the authors seem to be less explicit in interpreting their data in this regard. While their interpretation of BacA being a ATP-driven facilitator of VitB12 conceptually makes perfect sense and can explain how this ABC transporter can mediate VitB12 transport downhill its concentration gradient in either direction, I still would wonder whether the authors can exclude the possibility that in one of the transport directions (e.g. the import direction?), BacA mediates uphill transport.

Our measurements of the initial rates of transport hint at the possibility that there is limited uphill transport possible in the import direction, because transport rates in the import direction are 2-3 fold higher than in the export direction (See Figures 2 and 3), which would lead to a small concentration difference at steady state in the absence of any sinks. We mention this possibility in lines 298-301: “Therefore, in the presence of physiological concentrations of cobalamin, operation of BacA_{Mm} could lead to a low level of accumulation, albeit much less what the hydrolysis of ATP could achieve if all free energy was used for accumulation.” However, as mentioned in the second comment from the reviewer, BtuF may act as a sink in the lumen of the proteoliposomes, thereby lowering the free cobalamin concentrations, and making it difficult to accurately determine the potential accumulation levels. The relatively high apparent K_M values of cobalamin for both directions (3.9 μ M import, 40 μ M export) makes it difficult to conclusively demonstrate transport against the concentration gradient in our fluorescence-based accumulation setup, as the encapsulated sensor concentration would need to be drastically increased. A conventional radioactive transport assay may be required to provide further conclusive evidence to address the question of whether BacA is capable of uphill transport.

- 2) Another potential technical caveat is the high affinity of the VitB12 sensor BtuF. I am sure, the authors considered the fact that BtuF acts as a sink, thus driving VitB12 transport into the direction of the sensor. Given the authors look here at facilitated transport (and do not claim uphill transport to happen), I do not think this is really problematic for the interpretation of the results. But nevertheless, a short note on this point in the paper might not harm.

We now mention this possibility in the discussion in lines 267-269: “While the high-affinity binding property of BtuF₄₈₈ on one hand provides sensitivity to the transport assay, it may on the other hand also act as a sink, making it difficult to demonstrate transport of cobalamin against its concentration gradient.”

- 3) The authors did not include any data on BacA that carries mutations to abrogate ATP hydrolysis. While I appreciate the fact that the authors have conducted the transport assays using ADP as control (and thereby established that ATP is needed for VitB12 transport mediated by BacA), it would nevertheless have been interesting to see how a Walker B E-to-Q mutant would have behaved in terms of transport.

We have previously shown that the walker B E-to-G mutant of BacA_{Mt} is no longer able to transport cobalamin *in vivo* in the import direction. Moreover, the protein was found to be kinetically-trapped in the NBD dimerized state by ATP picked up during protein expression¹. We therefore speculate that it is unlikely that the protein is able to transport cobalamin in either direction *in vitro*, but it will be worthwhile to conduct a comprehensive mutagenesis study in the future, also targeting other conserved motifs

For now, we have included an additional experiment in which we used the non-hydrolysable ATP analogue AMP-PNP instead of ATP. We tested the ability of BacA_{Mm} to transport cobalamin from trans-to-cis (import direction) in presence of AMP-PNP using the release setup (Figure below, and updated Fig 2a). We mention this experiment in lines 161-164: “We did not observe quenching of the sensor in presence of external Mg-ADP (white triangles, Fig. 2a) or the non-hydrolysable ATP analogue AMP-PNP (grey circles, Fig. 2a), demonstrating that the transport of cobalamin is strictly coupled to the hydrolysis of ATP.”

[Minor points]

- 1) Line 140: I was surprised to see such a strong quenching in the setup wherein VitB12 was entrapped in the (tiny) liposome lumen and transported into the bulk solution where BtuF was present, which has a much larger volume. Did the author calculate/determine the maximal quenching of BtuF that can be achieved in this experimental setup?

In our setup, we make use of a total of 0.5 mg lipids. We prepared liposomes with a diameter of ~400 nm and encapsulated 13 μM of cobalamin inside the lumen of the liposomes. Hence, we have the following amount molecules:

- 0.5 mg lipids / 400 nm size = $2.65 \cdot 10^{11}$ liposomes, which encapsules $7.04 \cdot 10^{13}$ cobalamin molecules at a concentration of 13 μM
- 25 nM BtuF₄₈₈ in 200 μl volume = 5 pmol = $3.01 \cdot 10^{12}$ molecules

Therefore, based on the numbers of cobalamin and sensor molecules, maximal quenching -perhaps intuitively surprisingly- can be reached.

- 2) Line 140: The legend says “The data were obtained from biological duplicates containing each technical triplicates. The data is presented as the mean with error bars indicating the standard deviation.” What remains unclear (here and also in further instances) how the standard deviation was calculated: did the authors simply take the 6 data points to compute mean and STDEV, or did they calculated the STDEVs for the two biological replicates separately and performed an error propagation calculation to arrive at the final STDEV?

The 6 datapoints from the biological repeat and technical triplicates were used directly to calculate the mean and the STDEV for the data represented in Figure 1 and 2. For Figure 3, the STDEV was calculated by taking the mean of the two averaged biological repeats. We now state explicitly how the STDEV was calculated in lines 141, 222 and 247.

- 3) Line 210: Author statement: “Again, a 2-3-fold higher transport rate was found in the import direction compared to the export direction, indicating that transport in the trans-to-cis (import) direction is somewhat more efficient than in the trans-to-cis (export) direction, and that the transporter had been reconstituted in equal proportions in the right-side-out and inside-out orientation.” It would have been nice/desirable, if the authors would have estimated the orientation of insertion of the BacA using an orthogonal method, e.g. tag-cleavage by a protease followed by Western blotting.

We believe that the way we calculated the ratio between orientations (by using protein activity) is superior to other methods, because we look at active protein only. In the orthogonal assay, inactive/misfolded protein may complicate the interpretation. In addition, activity-based assays -in contrast to the orthogonal assay- are not confounded by the potential presence of multilamellar liposomes. Therefore, we prefer to stick to our activity-based results.

- 4) Line 218: “10 mM externa Mg-ATP (blue circles),...” -> typo: external

We corrected the typo.

Reviewer #2 (Remarks to the Author):

The study by Nijland et al. characterizes the direction of transport of the ABC transporter BacA. According to the current paradigm, ABC transporters are unidirectional, i.e. they can either import or export substrates, but never transport in more than one direction. This distinguishes ABC transporters from secondary transporters, which are bidirectional by definition. Here, the authors propose that BacA is indeed an ABC transporter that allows transport in two directions. This is an extraordinary claim, and the authors are aware that it requires extraordinary evidence.

The experimental design and its execution are very strong and result in convincing and complete data sets. For example, my concern that the reduced transport rates (Fig. 2, BacA transport) could be explained by a reduced internal concentration of ATP and/or cobalamin due to ATP hydrolysis or cobalamin export during sample preparation was properly addressed by monitoring transport in the opposite direction as well.

Therefore, I can only share the authors' conviction that they have measured bidirectional transport of the ABC transporter BacA, which is a sensational finding.

Major comment

- Line 133, 'co-encapsulated'. The ability to export cobalamin from cis to trans depends on the presence of ATP. Clearly, ATP is co-encapsulated. However, 1) several ABC transporters show uncoupled ATP hydrolysis; and 2) ATP hydrolysis is even more likely to occur in the presence of a substrate. If this is the case for ECF-CbrT, this could explain the lack of transport in the export direction: either no ATP is left or all substrate has already left the liposome due to cis-to-trans export activity during sample preparation. Please explain why this scenario is not considered.**

We agree and appreciate the worry that many ABC transporters have basal uncoupled ATPase activity, which could certainly interfere with the experimental setup. In this case we are convinced that the lack of export activity reported for ECF-CbrT in our release setup is not caused by the depleted luminal ATP or cobalamin, as we will explain below. We also would like to stress that we used ECF-CbrT in the current work only to develop the assay. The bidirectional transport observed for BacA is the primary message of our work, and this conclusion is not dependent on the ECF-CbrT mechanism:

- We can exclude that all cobalamin substrate has been transported out of the liposomes during the sample preparation. While we are not able to visualize transport in the cis-to-trans transport activity in presence of luminal ATP (blue trace, Fig. 1b and Supplementary Fig. 3), we do observe release of cobalamin when ATP is *also* added on the outside of the same sample preparation (See purple trace, Supplementary Fig. 3). While the quenching rate of the sensor is slower compared to the sample where ATP is only added externally (which we speculate to happen because of the re-transport of cobalamin into the lumen, see main text), this proves that there is still cobalamin in the lumen of the liposomes.
- When the proteoliposomes are disrupted with 0.15% triton X-100 in the end of the experiment, we observe significant quenching of the sensor, indicating that cobalamin is released into the external buffer (See Fig. below).

- To additionally support our interpretation, we have now tested the ability of ECF-CbrT to transport cobalamin in the trans-to-cis (import) and cis-to-trans (export) direction using the inverted (accumulation) setup. In this setup, we observe quenching of the sensor in presence of luminal Mg-ATP (Yellow circles, Fig. below, and new Fig. 1c), confirming that ECF-CbrT transports cobalamin in the import direction. This demonstrates that sufficient ATP molecules remain present in the lumen of the liposomes during sample preparation to support transport of cobalamin. In stark contrast to BacA_{Mm}, we did not observe quenching of the sensor when Mg-ATP and cobalamin were both added to the external buffer (blue circles, Fig. below), again confirming that ECF-CbrT lacks the ability to transport cobalamin in the cis-to-trans (export) direction. In this setup, the lack of export activity cannot be caused by the depletion of ATP or the absence of substrate, because both are added at the last moment to trigger the transport process.

Minor comments

- The authors have specifically designed their assay to determine the direction of transport. Please comment on the ability of more conventional transport experiments (e.g. those using radioactive substrate) to reveal (or hide) bidirectional transport. What are the chances that this is a more common feature that has gone undetected due to a bias in the experimental system? In this regard, please explain why ECF-CbrT (line 111) can be considered “strictly unidirectional” based on ref28 and 29.

We thank the reviewer for bringing up this point. Indeed, our experiments show that the experimental setup is crucial to detect bidirectional transport. The ability of BacA to transport cobalamin from cis-to-trans would have remained unexplored if we had excluded the experiment where both ATP and substrate were added to the same side of the liposomal membrane. While our fluorescence-based setup made it more straightforward to detect transport in both directions (cis-to-trans and trans-to-cis), conventional radioactive transport assays should also support these type of measurements when the nucleotides are added on the other side of the membrane. Instead, the assumption that ABC transporters exploit strictly unidirectional transport has generally restricted the standard experimental setup of transport assays to visualize transport in only a single direction. We now add a statement making this point clearer (lines 287-289: “It is possible that bidirectionality may be found in more ABC transporters, but has so far remained undetected because the assay conditions prevented detection.”

We agree that references 28 and 29 do not conclusively show that ECF-CbrT operates in a strictly unidirectional fashion, and have rephrased accordingly.

- Line 172, “transport [...] export direction”. Needless to say, any remaining external ATP from the co-encapsulation procedure would also have led to the observation of cobalamin leaving the proteoliposomes. However, the author followed a procedure of washing and resuspension that makes it unlikely that any external ATP is present. The authors should emphasize this in their text.

We added text to emphasize this point (lines 179-181).

- 4) Line 229, the affinity is clearly different. I am concerned about the three digit precision for transport in the export direction, which seems unrealistically high. This value assumes that transport is saturated, but it is not clear from fig3a that this is indeed the case for the export direction. Instead, I suggest that the authors rephrase to indicate that the K_M -app is at most 38 μM .

We rephrased accordingly (lines 231-234)

- 5) Line 234, "less than twofold" -> what data are the authors referring to? Is this panel 3A? If so, please confirm that the same V_{max} value was used for normalization.

We indeed used non-normalized transport rates to calculate V_{max} (source data file related to Figure 3) In addition, we also calculated the V_{max} values based on the data in Fig 2a/b (combined with the K_M values from fig 3). There are small differences when V_{max} values are derived from the different experiments (as is very common for proteoliposome-based transport assays,) but the differences between the V_{max} for import and export are always less than two-fold.

- 6) Fig. 3b, please indicate at what cobalamin concentration these measurements were made.

We added the concentrations of B12 and Mg-ATP to the legend of Fig. 3a.

- 7) Line 60, "smuggles" suggests some degree of secrecy. An alternative would be to indicate that toxic compounds are taken up as part of a moonlighting activity of the protein.

We rephrased accordingly.

- 8) Line 168: import should be trans-to-cis

We corrected the mistake.

- 9) Line 219: externa -> external

We corrected the typo.

- 10) Line 230: "only a fraction" -> please explain this statement. It is not clear why the authors assume that only a fraction of the sensors are accessible. Is this due to the expected heterogeneity in the distribution of the sensors across the liposomes? Or has this been demonstrated (where?)?

We extended the statement (lines 234-238). The reviewer is correct that heterogeneity in the liposome population leads to an inaccessible fraction, either because of multi-lamellarity, or because of a fraction of liposomes without active reconstituted transport protein.

- 11) Line 245: The first paragraph does not seem to be relevant to this manuscript and I suggest deleting it.

We had discussions between the authors on whether or not to include this part from the very initial stages of the manuscript, and back then decided to keep it, because it provides a rationale for the way we set up the experiments. Therefore, we prefer to keep it, but leave it to the editor to decide.

References

1. Rempel, S. *et al.* A mycobacterial ABC transporter mediates the uptake of hydrophilic compounds. *Nature* **580**, 409–412 (2020).

Reviewer #1 (Remarks to the Author):

The authors have carefully addressed all points raised by me and the other reviewer in a very convincing manner.

In particular, I was convinced by the authors' argument that performing orthogonal experiments showing the directionality of transporters reconstituted into the liposomes by proteases are inferior to their own analysis based on transport only, because in transport experiments, one only "sees" the active proportion of transporters (i.e. I learned something...).

I wish to congratulate the authors for this nice and neat story once more!

Reviewer #2 (Remarks to the Author):

The authors have appropriately addressed my concerns.

I do note that the cartoon accompanying fig 1c appears to have been accidentally swapped: the cartoon for the yellow trace is next to the blue trace, and vice versa.

REVIEWERS' COMMENTS

Reviewer #1 (Remarks to the Author):

The authors have carefully addressed all points raised by me and the other reviewer in a very convincing manner.

In particular, I was convinced by the authors' argument that performing orthogonal experiments showing the directionality of transporters reconstituted into the liposomes by proteases are inferior to their own analysis based on transport only, because in transport experiments, one only "sees" the active proportion of transporters (i.e. I learned something...).

I wish to congratulate the authors for this nice and neat story once more!

We thank the reviewers once more for the flattering comments

Reviewer #2 (Remarks to the Author):

The authors have appropriately addressed my concerns.
I do note that the cartoon accompanying fig 1c appears to have been accidentally swapped: the cartoon for the yellow trace is next to the blue trace, and vice versa.

We thank the reviewer for noticing, and have now swapped the cartoons